# OpenReview forum: "SafeCoop: Unravelling Full Stack Safety in Agentic Cooperative Driving"
_ICLR.cc/2026/Conference — ICLR 2026 Conference Withdrawn Submission_

### Official Review · Reviewer_M1ko · 2025-10-28

**Soundness:** 3
**Presentation:** 4
**Contribution:** 2
**Rating:** 6
**Confidence:** 4

**Summary:**

This paper presents a comprehensive study on full-stack safety and security issues in natural-language-based collaborative driving. The authors develop a taxonomy covering both generic threats such as connection disruption and relay interference, and language-specific vulnerabilities like content spoofing. They introduce SafeCoop, a defense pipeline integrating a semantic firewall, language-perception consistency checks, and multi-source consensus. The experimental evaluation in CARLA simulations demonstrates the approach's effectiveness against adversarial scenarios. While the methodology is well-structured and the analysis thorough, the defense mechanisms primarily adapt existing natural language processing security techniques rather than introducing novel multi-agent security innovations. The paper's key contribution lies in systematically addressing security gaps for language-mediated vehicular cooperation, though it does not sufficiently explore emergent security challenges arising from multi-agent coordination dynamics in adversarial environments. This work provides valuable insights for robust defense design in natural language-based V2X systems despite the limited novelty in its core defense architecture.

**Strengths:**

1. It develops a well-structured taxonomy that effectively categorizes both generic threats and language-specific vulnerabilities.

2. The methodology is clearly presented with a thorough analysis of the proposed solution.

3. The work systematically addresses security gaps in language-mediated vehicular cooperation frameworks.

4. The SafeCoop defense pipeline offers practical insights for building robust natural language-based V2X systems.

**Weaknesses:**

1. Some of the proposed attacks, such as connection disruption and relay interference, are not specific to natural language-based collaborative driving systems.

2. The defense mechanisms primarily adapt existing natural language processing security techniques rather than introducing novel multi-agent security innovations.

3. The work fails to adequately highlight how the multi-agent perspective creates unique security considerations not addressed in standard NLP security approaches.

**Questions:**

1. From a multi-agent perspective, what aspects of the proposed attacks and defenses most clearly distinguish this work from existing NLP security research? Specifically, how do the interactions between agents in collaborative perception introduce unique vulnerabilities that traditional NLP security approaches do not address? Does the SafeCoop framework effectively mitigate these through its semantic firewall language-perception consistency checks and multi-source consensus mechanisms?

2. What assumptions does the defense mechanism make regarding the ego vehicle's knowledge of the attacker's capabilities? How does the proposed defense perform against unknown or adaptive attacks?

3. How does the language-driven driving system meet the real-time requirements of autonomous vehicles? What is the computational overhead introduced by the additional defense components, like semantic firewall analysis and language-perception consistency verification, and could these impact critical driving decisions in time-sensitive scenarios?

---

### Official Review · Reviewer_TG64 · 2025-10-30

**Soundness:** 3
**Presentation:** 2
**Contribution:** 2
**Rating:** 2
**Confidence:** 4

**Summary:**

This paper presents the first systematic study of safety in natural-language-based collaborative autonomous driving, identifying four attack surfaces (Connection Disruption, Relay/Replay Interference, Content Spoofing, and Multi-Connection Forgery) and proposing SafeCoop, an agentic defense pipeline with three specialized agents. Evaluated on 32 CARLA scenarios, SafeCoop achieves 69.15% driving score improvement under attacks and 67.32% F1 score for malicious detection.

**Strengths:**

+ Novel and Timely Problem Formulation

+Comprehensive Attack Taxonomy

+ Closed-loop and Thorough Evaluation

**Weaknesses:**

Weakness

- Unclear relationship between dual objectives: The system outputs a trust score for each agent, but claims to address both "performance" and "anomaly detection" objectives. However, the system design only explicitly addresses anomaly detection through trust scoring and filtering. It remains unclear how performance is directly optimized, or whether the authors simply assume that anomaly detection will inherently improve performance. This assumption should be explicitly stated and justified.

- Insufficient attack implementation details: While Section 3 and Appendix D provide a taxonomy of attacks, the paper lacks a detailed explanation of how attacks are actually designed and implemented in the evaluation. For reproducibility and proper assessment of the defense mechanisms, more specifics are needed on attack generation, particularly for the MLLM-based Content Spoofing attacks (e.g., prompt engineering strategies, attack success rates, stealthiness measures). Further, in Table 1, several adversarial scenarios (w/o defense) achieve better performance than the Benign (Non-collab) baseline across multiple metrics. This counterintuitive result raises questions about attack severity and undermines claims about the effectiveness of the defense. While Appendix G partially addresses this through the computation scaling phenomenon, this critical finding deserves prominent discussion in the main paper with a thorough analysis of what it implies for both attack potency and defense necessity.

- Unexplained performance degradation in ablation study: Table 3 shows that adding LPC and MSC agents actually increases the Vehicle Collision (VC) score under CS+MCF attacks, which is concerning. The paper provides no investigation or explanation for this degradation. Could false positives in anomaly detection be causing the defense to filter legitimate messages, thereby reducing situational awareness and increasing collisions? Without presenting false positive/false negative rates and analyzing this phenomenon, the reliability of the defense pipeline remains questionable.

**Questions:**

How is "performance" explicitly optimized in the system design, or is it assumed to naturally follow from anomaly detection?

Why do some adversarial scenarios (Table 1) outperform the non-collaborative baseline? What does this imply about attack severity?

What causes LPC and MSC agents to increase vehicle collisions under CS+MCF attacks (Table 3)? Are false positives responsible?

Why does the Firewall Agent require MLLMs for semantic reasoning when it appears to only process JSON-formatted input?

---

### Official Review · Reviewer_4Rwz · 2025-11-02

**Soundness:** 2
**Presentation:** 3
**Contribution:** 2
**Rating:** 2
**Confidence:** 4

**Summary:**

This paper proposes using LLMs to safeguard natural language-based cooperation among vehicles. The paper summarizes the taxonomy of the attack surface, which includes Connection disruption, Relay interference, Content spoofing, and Sybil attack (multi-connection forgery). The proposed approach combines several LLM agents, each more effective in one, or a subgroup of, defense. Evaluation using Carla shows defense performance using end-to-end metrics such as driving score, route completion, etc. The ablation study showcases different agents' capabilities, and the authors also compare the performance among multiple LLMs as the defense agents.

**Strengths:**

+ The paper explores cooperative driving security in language domain.
+ The writing quality is above average though missing a lof of details and clarity
+ The evaluations are systematic and comprehensive

**Weaknesses:**

- Missing details on the range/magnitude of attack being applied during the evaluation. How is each of the attacks staged? What is the range of the attack, e.g., How much temporal misalignment in relay/replay interference? What is being introduced in content spoofing? How many forged connections are there in the Sybil attack? These are important details to provide context to gauge the evaluation metrics. Whether these staged attacks are trivial or sophisticated, the results can lead to entirely different conclusions. One way to show the sophistication is to show if conventional defense against these attacks can be successful, how does LLM compare against conventional methods, and whether using an LLM is an overkill.
- Missing details on how the LPC agent compares ego perception with received messages. Is it taking a multi-view image or lidar as input? What if the received message is outside the field of view of the perception module?
- Figure 1, appearing before the abstract, could use additional legends and details to be self-explanatory. What do the numbers after CD, CS, and CS+MCF mean? Why is the MCF description pointing to the CS score? And CS description points to CS+MCF score? Before the readers read the abstract, a couple of legends and explanations could be very helpful to avoid confusion.
- The paper claims to have studied four attack surfaces (CD, RI, CS, MCF). What has been studied for CD (connection disruption)? There are no results on CD in Table 1. Table 2 is attack detection, not defense. If it's difficult to detect and evaluate, ok to tone down a bit.
- The evaluation results show LLMs are insufficient in safeguarding against attacks, cannot recover collaborative driving performance before attacks, and cannot be run in real-time. These observations are good to know, but marginally advance the field's understanding. It would be helpful to show contributions/insights obtained to advance the defense.

**Questions:**

Please refer to weakness section

---

### Official Review · Reviewer_NsNL · 2025-11-02

**Soundness:** 3
**Presentation:** 3
**Contribution:** 2
**Rating:** 4
**Confidence:** 3

**Summary:**

This paper studies full-stack safety for natural-language-based collaborative driving systems. It identifies four attack surfaces (connection disruption, relay/replay, content spoofing, and multi-connection forgery) and proposes SafeCoop, an agentic defense pipeline combining (i) semantic firewalling, (ii) language-perception consistency checking, and (iii) multi-source consensus with temporal checks.

Evaluations in CARLA across 32 scenarios demonstrate that malicious language communication severely harms collaborative driving, and SafeCoop substantially recovers performance and detects adversarial agents, achieving up to ~69% driving score improvement and ~67% F1 detection.

**Strengths:**

1. Timely problem: Addresses emerging risks in language-based V2X collaboration, an important but under-explored area as driving LLMs become more capable.

2. Comprehensive threat model: The taxonomy spans channel-level and semantic attack vectors, grounding them in adversarial V2X literature.

3. Agentic defense pipeline: Novel multi-module agent approach (firewall, perception-language consistency, consensus), offering interpretability.

**Weaknesses:**

1. Novelty feels incremental.
The paper mainly repackages known security concepts (trust-based filtering, majority consensus, temporal consistency) into an LLM-driving setting.
While the agentic framing is interesting, the core mechanics resemble classical V2X trust scoring + consistency checks, raising questions about conceptual novelty.

2. Limited realism & scalability assumptions.
The system assumes synchronous simulation, and perfect ego perception during consistency checks.
Real V2X networks are asynchronous, lossy, and perception-noisy, which may reduce effectiveness.

3. High latency & unclear deployment feasibility.
Even fast models are around 700ms and larger models exceed 3s latency, far above real-time driving constraints.
The paper acknowledges this but does not meaningfully address deployment pathways.

4. How frequently is the defense executed?
If the defense must run continuously or per-frame, the computational burden may be prohibitive; a one-time malicious-agent flag is insufficient for practical systems. Clarifying the defense invocation frequency and cumulative runtime overhead (e.g., cost per second of driving) would be important to judge real-world viability.

5. Formula Clarification. For Eq(2), why is the attacker constrained to modifying a single sender? Additionally, the expression for $D_j$ appears to use observation $o_i$ instead of $o_j$. Is it intentional or an indexing error?

**Questions:**

Please respond to each point in Weaknesses.

---

### Note · Authors · 2025-11-13

I have read and agree with the venue's withdrawal policy on behalf of myself and my co-authors.